# Effects of Neuromuscular Training on Postural Balance and Physical Performance in Older Women: Randomized Controlled Trial

**DOI:** 10.3390/jfmk9040195

**Published:** 2024-10-14

**Authors:** Yeny Concha-Cisternas, José Castro-Piñero, Manuel Vásquez-Muñoz, Iván Molina-Márquez, Jaime Vásquez-Gómez, Eduardo Guzmán-Muñoz

**Affiliations:** 1Escuela de Kinesiología, Facultad de Salud, Universidad Santo Tomás, Talca 3460000, Chile; yenyconchaci@santotomas.cl; 2Vicerrectoría de Investigación e Innovación, Universidad Arturo Prat, Iquique 1100000, Chile; 3GALENO Research Group, Department of Physical Education, Faculty of Education Sciences, University of Cádiz, 11519 Puerto Real, Spain; 4Instituto de Investigación e Innovación Biomédica de Cádiz (INiBICA), University of Cádiz, 11009 Cádiz, Spain; 5Centro de Observación y Análisis de Datos en Salud, Facultad de Medicina y Ciencias de la Salud, Universidad Mayor, Santiago 8580745, Chile; 6Escuela de Medicina, Facultad de Medicina y Ciencias de la Salud, Universidad Mayor, Santiago 8580745, Chile; 7Escuela de Educación Física, Facultad de Educación, Universidad Adventista de Chile, Chillán 3780000, Chile; 8Centro de Investigación de Estudios Avanzados del Maule (CIEAM), Universidad Católica del Maule, Talca 3460000, Chile; 9Laboratorio de Rendimiento Humano, Universidad Católica del Maule, Talca 3460000, Chile; 10Pedagogía en Educación Física, Facultad de Educación, Universidad Autónoma de Chile, Talca 3460000, Chile

**Keywords:** aging, neuromuscular training, postural balance, physical performance, muscle strength

## Abstract

**Background:** Aging causes morphological and physiological changes, especially in the musculoskeletal and somatosensory systems, leading to sarcopenia and reduced postural balance, increasing the risk of falls and loss of independence. This study aimed to analyze the effects of a neuromuscular training program on postural balance, physical performance, and muscle strength in older Chilean women. **Methods:** The double-blind randomized controlled trial included 48 participants aged 65–80 who were divided into three groups: a control group (CG), a multicomponent training group (MCG), and a neuromuscular training group (NMG). Postural balance was assessed using a force platform with open eyes (OE) and closed eyes (CE) conditions, measuring ML velocity, AP velocity, mean radius, and area. Physical performance was evaluated with the Short Physical Performance Battery (SPPB), including static balance, 4-m walking speed, and 5× sit-to-stand chair test. Muscle strength was measured using a hydraulic hand dynamometer to record maximum grip force. **Results:** Significant time × group interactions in the CE condition were found for mean radius (F = 0.984; *p* = 0.017; ηp^2^ = 0.184), AP velocity (F = 10.01; *p* = 0.001; ηp^2^ = 0.312), and ML velocity (F = 4.30; *p* = 0.027; ηp^2^ = 0.163). Significant pre–post differences in the NMG for mean radius (*p* < 0.001), AP velocity (*p* < 0.001), and ML velocity (*p* = 0.029) were observed, with no significant changes in CG. Significant time × group interactions were also found in the SPPB test score (F = 11.49; *p* < 0.001; ηp^2^ = 0.343), gait speed (F = 5.513; *p* = 0.012; ηp^2^ = 0.198), and sit-to-stand test (F = 5.731; *p* = 0.010; ηp^2^ = 0.206), but not in the balance score (F = 2.099; *p* = 0.148). Handgrip strength showed no significant interactions (F = 3.216; *p* = 0.061; ηp^2^ = 0.127). **Conclusions**: These findings suggest that neuromuscular training is a promising intervention to mitigate the decline in balance and physical function associated with aging, offering a targeted approach to improve the quality of life in the elderly.

## 1. Introduction

Population aging can be considered one of the most significant social transformations of the 21st century [1]. Globally, in 2015, there were 901 million people aged 60 or older, while by 2050, the number is projected to be at least 2.1 billion people aged 60 or older [1]. A similar situation is occurring in Chile, where 18.1% of the population was elderly in 2022, and this population is expected to reach 32.1% by 2050 [2].

During aging, morphological and physiological changes occur in all body systems [3]. However, the changes that affect the musculoskeletal and somatosensory systems are highly frequent in the elderly population [3]. At the musculoskeletal level, the development of sarcopenia is observed as a syndrome characterized by loss of muscle mass and function, muscle strength, and performance [4,5]. This decrease in muscle mass compromises the ability to perform daily activities, such as getting up from a chair, walking, or climbing stairs, which results in a deterioration of functional independence and a greater susceptibility to falls and injuries.

In addition, with advancing age, there is a deterioration of the components of the somatosensory system, which includes sensory receptors located in the skin, tendons, muscles, and ligaments [6]. The literature indicates that aging leads to a reduction in the quantity and density of mechanoreceptors such as the Golgi tendon organ and the muscle spindle, which play a fundamental role in kinesthesia (i.e., the perception of movement), the sensation of joint position, and the control of muscle tension [7]. These changes affect the sensation, perception, and execution of body movements, altering various motor functions, such as static and dynamic postural balance [8,9].

Postural balance refers to the ability to control or maintain the posture of the entire body or its center of mass in relation to the base of support [10]. Henry and Baudry found that older adults exhibit alterations in postural balance, which manifest as increased body swaying, greater coactivation of antagonist and agonist muscles, higher dependence on visual information, and reduced automatic control of an upright position [11]. Similarly, studies led by Lin and Woollacott reported that postural balance in older adults is affected both by a reduction in sensory integration and by a decline in muscle strength [12]. This finding was confirmed by Yu Kim et al., who indicated that sarcopenia increases the risk of developing postural balance difficulties by 1.7 times, regardless of age and sex [13].

Currently, training methods have been recommended to counteract and improve the negative effects of aging [14,15,16]. Among them, multicomponent training has established itself as a methodology with solid evidence, as it has been shown to improve physical and mental aspects in the elderly population [16,17,18,19]. However, this type of training includes motor activities that often limit the individualized adaptation of exercises to the specific capabilities and needs of each person. In contrast, shorter rehabilitation training sessions that include sensory stimuli along with motor stimuli, such as neuromuscular training, are still rarely addressed in older adults [20,21]. The concept of neuromuscular training describes the combination of proprioceptive muscle strength and postural balance exercises as part of a comprehensive rehabilitation program [22]. This intervention is based on providing adequate information to peripheral receptors, specifically mechanoreceptors, so that the integration of muscle responses is more efficient [23]. Previous studies have shown that a 4-week neuromuscular training program improved postural control in collegiate volleyball players with functional ankle instability [24]; meanwhile, Orellana et al. demonstrated that a 6-week neuromuscular training program improved dynamic postural balance and proprioception in female basketball players [25]. In older adults, an international study showed that an 8-week neuromuscular training program improved postural balance variables in older adults with diabetic neuropathy [26]. In Chile, the literature on older adults is still limited, with only one pre-experimental study on individuals with knee osteoarthritis showing improvements in postural balance and functionality after an 8-week intervention [27]. Considering the advanced aging process of the Chilean population and the negative effects that aging has on postural balance, physical function, and muscle strength, this study aimed to analyze the effects of a neuromuscular training program compared to a multicomponent training program on postural balance, physical performance, and muscle strength in older Chilean women. We hypothesized that neuromuscular training would have a more significant effect on postural balance, physical performance, and muscle strength than multicomponent training alone compared to a control.

## 2. Materials and Methods

### 2.1. Study Design

This study corresponds to a double-blind (participants and physical therapists) randomized controlled clinical trial that presents three parallel groups: control group (CG), multicomponent group (MCG), and neuromuscular group (NMG). A research randomization website was used for randomization (https://www.randomizer.org, accessed on 30 March 2022). The methodology followed was the Consolidated Standards of Reporting Trials Statement (CONSORT) guidelines [28]. All participants voluntarily read and signed an informed assent while their legal guardians authorized participation by signing an informed consent. The research was approved by the ethics committee of the Universidad Santo Tomás, Chile (No. 115-21) and was based on the Declaration of Helsinki.

### 2.2. Participants

The sample size was calculated using Gpower software version 3.1.9.6 [29] based on the mean difference previously reported in a clinical trial of the neuromuscular program in older people [26]. The study indicated that a minimum mean difference of 0.23 mm in anteroposterior and center of pressure (COP) sway is needed to achieve meaningful clinical improvements. A sample size calculation was performed to determine this, requiring at least 13 participants per group with a significance level of 0.05 and a power of 80%. Initially, 59 older women were recruited from five community centers in Talca, Chile. Of these, some were excluded due to failing to meet the eligibility criteria. As a result, 48 older women were included and randomly assigned to three groups of 16 participants each to account for potential dropouts during the intervention. The criteria for inclusion were: (i) women aged between 65 and 80 years; (ii) those with a functional status indicating either self-sufficiency or risk of self-sufficiency as measured by the Older People Functional Assessment Measure (EFAM-Chile) [30]; (iii) individuals achieving a score of 14 or higher on the Mini-Mental State Examination, reflecting adequate cognitive function [31]; and (iv) those who had received medical clearance to engage in physical activity. The exclusion criteria were: (i) individuals with a suspected cognitive impairment or difficulty understanding instructions; (ii) those with cardiovascular, respiratory, or other medical conditions that would impede their participation in a physical activity; (iii) participants diagnosed with a vestibular disorder by a healthcare professional; (iv) individuals who had undergone surgery in the six months before the study; (v) those with an uncontrolled chronic illness; and (vi) those who were participating in another physical activity or rehabilitation program.

### 2.3. Randomization and Blinding

A random number sequence was generated using www.random.org, and a block randomization strategy with variable block sizes was employed to ensure an equal and unpredictable distribution of participants across the control, multicomponent training, and neuromuscular training groups. To maintain allocation concealment, sequentially numbered, opaque, sealed, and stapled envelopes were used, ensuring the integrity of the randomization process. All physical therapists were blinded to the randomization process, not knowing the allocation list or the existence of other training groups. Participants were also blinded to their group assignments and instructed not to disclose any details regarding their group or activities. Additionally, interventions were conducted at separate locations to prevent cross-contamination between groups. Despite these measures, full blinding of all assessors was challenging due to the nature of some assessments used in the study.

### 2.4. Outcomes

The study variables were assessed in the biomechanics laboratory at the School of Kinesiology at Santo Tomás University, Chile. To describe the sample, body weight, height, and body mass index (BMI) were measured. Weight and height were recorded using a SECA 700 mechanical scale with an integrated stadiometer (Hamburg, Germany; precision of 0.1 kg and 0.1 cm). Waist circumference (WC) was measured using a tape measure at the narrowest part of the torso located between the lowest rib and the iliac crests at the end of expiration while standing [32]. Subsequently, BMI was computed using internationally accepted criteria by dividing body weight by the square of height (kg/m^2^) [32].

#### 2.4.1. Postural Balance

Postural balance was assessed using a force platform (Artoficio model 0813, Santiago, Chile) that was 40 × 40 cm in size. Postural balance was measured with open eyes (OE) and closed eyes (CE) conditions, each lasting 30 s. Participants were instructed to stand on the platform and maintain a bipedal position with their arms relaxed at their sides, their feet shoulder-width apart, and their OE looking at a fixed point [33]. For this assessment, the participants had to be barefoot. Three trials were performed and averaged to obtain the postural balance and COP variables: mediolateral (ML) velocity, anteroposterior (AP) velocity, mean velocity, mean radius, and area. A higher value of these variables indicates poorer postural balance. The test was then repeated with CE, restricting visual feedback using a blindfold. The data were acquired at a sampling rate of 40 Hz.

#### 2.4.2. Physical Performance

Physical performance was assessed through the Short Physical Performance Battery (SPPB) [34], which consists of 3 tests. The first one evaluates static balance through 3 positions. First, the elderly person was asked to stand and maintain a “feet together” position, then continued with the “semi-tandem” position, and finished with the “tandem” position. In all 3 cases, the time that the user could maintain these positions was timed [35,36]. The second test was the 4-m walking speed test. For this test, the participant had to remain seated, and upon hearing the instructions, she had to stand up and walk at her usual pace for a previously delimited distance of 4 m. The execution time was timed (Casio^®^ HS-3 model stopwatch, Tokyo, Japan). Finally, the 5× sit-to-stand chair test was performed. The participant had to remain seated in a chair without armrests and, upon instruction, had to stand up and sit down 5 times as quickly as possible. The time that the user took to complete the test was timed [17,36]. The total score, considering the 3 tests of this battery, ranges from 0 points (poor performance or low functional performance) to 12 points (optimal performance or high functional performance).

#### 2.4.3. Muscle Strength

Muscle grip strength was assessed via the manual grip strength test using a hydraulic hand dynamometer (Jamar model 5030 J1, Sammons Preston Rolyan, Bolingbrook, IL, USA) that was previously calibrated [34]. To perform the test, participants were seated with their shoulders adducted and elbows flexed at 90°, with forearms and wrists in a neutral position. The segment being evaluated had no external support. The dynamometer was positioned vertically. Participants were instructed to exert maximum grip force with their dominant hand for 3 s per attempt with a 1-min rest between each repetition. Each participant completed three attempts [37,38]. The average of the attempts was recorded in kilograms.

### 2.5. Interventions

#### 2.5.1. Multicomponent Training Program

The multicomponent training program was designed based on physical activity guidelines for older adults as outlined in the “Vivifrail” recommendations [17,39]. The program included exercises targeting endurance, strength, flexibility, and balance–agility. Participants attended two 90-min sessions per week on non-consecutive days over a 12-week period, totaling 24 sessions. Each session included progressive increases in intensity and repetitions for the strength and muscular endurance exercises, with intensity progression monitored using the Borg scale and the talk test. The program targeted major muscle groups in both the upper and lower body, utilizing dumbbells and elastic bands of varying resistance levels. During the first 3 weeks, light dumbbells (0.5–1 kg) and low-resistance bands (yellow) were used. Exercises included seated biceps and triceps curls, shoulder flexion, and abductions with elastic bands. Additionally, participants performed squats and sit-to-stand movements from a chair while holding weights. Three sets of 8–10 repetitions were prescribed. In the next 3-week block (weeks 4–6), the dumbbell weight was increased to 2–3 kg, and medium-resistance bands (green) were introduced. The exercises from the previous phase were maintained but executed with higher loads. New exercises were added, such as seated knee extensions and hip abductions with elastic bands, as well as supported half-squats. Three sets of 12–15 repetitions were prescribed. During the third block (weeks 7–9), dumbbell weights were adjusted to 3–4 kg, and high-resistance bands (blue) were used. Exercises from earlier phases were continued, with the addition of shoulder elevation in the scapular plane with elastic bands, half-squats with dumbbells, and diagonal movements with dumbbells. Four sets of 8–10 repetitions were prescribed. Finally, in the last 3 weeks (weeks 10–12), 5 kg dumbbells and maximum-resistance bands were employed. Participants performed some of the exercises from the previous phases with increased loads, and new exercises were introduced, including squats with dumbbells and a combination sequence: sit-to-stand + shoulder flexion with dumbbells, and lateral pulldowns with elastic bands. Four sets of 12–15 repetitions were prescribed. Throughout the 12 weeks, balance and agility exercises were also incorporated. The progression included activities utilizing various bases of support and positions: feet separated, feet together, semi-tandem, and tandem, gradually advancing to single-leg activities. The activities were conducted by a physiotherapist with expertise in exercise prescription.

#### 2.5.2. Neuromuscular Training Program

The neuromuscular training program was added to the multicomponent training for 12 weeks, distributed in 2 sessions per week on non-consecutive days for a total of 24 sessions. First, the multicomponent training program was performed with a duration of 60 min, and then 30 min of neuromuscular training was added, totaling 90 min per session. The neuromuscular training program included progressive upper and lower limb work, proprioception, strength, and balance exercises (Figure 1 and Figure 2). During weeks 1 to 3, exercises were performed on a stable surface with eyes open, including maintaining various foot positions (separated, together, semi-tandem, and tandem) (Figure 1A,B). Additionally, participants had to walk in a straight line (Figure 1C) and in a zigzag pattern on a coordination ladder without losing balance; during weeks 4 to 6, the exercises became more complex by incorporating the challenge of closed eyes while performing them on a stable surface. In weeks 7 to 9, the exercises were further advanced by introducing an unstable surface, performed barefoot and with eyes open, adapting the previous tasks to this new context (Figure 1D,E). Static walking with knee elevation was also added (Figure 1F). Finally, weeks 10 to 12 culminated with the execution of exercises on unstable surfaces, both with eyes open and eyes closed (Figure 1G), and with a reduction in the support surface (Figure 1H,I). The upper limb activities were also progressive. During the first 3 weeks, precision exercises were performed against a wall using a ball. The participant had to trace circles of different sizes on the wall (Figure 2A), gradually progressing to more complex patterns with smaller balls (Figure 2B). During weeks 4 to 6, participants were required to repeat the activity with their eyes closed. In weeks 7 to 9, the progression included tracing patterns on the wall with balls while in a single-leg stance and tracing from a greater distance using a laser pointer (Figure 2C,D). Similarly, during weeks 10 to 12, precision activities were incorporated by throwing balls of different sizes, modifying the throwing distance and targets (Figure 2E,F).

#### 2.5.3. Control Group (CG)

These participants received no intervention and were instructed to continue their daily activities and lifestyles. After completing the training and all assessments, they were offered the opportunity to engage in a 12-week neuromuscular training program as compensation.

### 2.6. Statistical Analysis

The statistical analysis was conducted using GraphPad Prism version 9.0 software. Descriptive statistics were computed, including the mean, standard deviation, and 95% confidence interval, to characterize the study results. An intention-to-treat (ITT) analysis was conducted to account for all participants as originally randomized, regardless of whether they completed the intervention. Missing data were addressed using a single imputation with the last observation carried forward (LOCF) method. This approach allowed for the estimation of missing values and ensured that the full sample size was maintained for the analysis. The ITT analysis was employed to provide conservative estimates of the intervention effects, accounting for potential biases due to attrition.

The Shapiro–Wilk test assessed normality, the Levene test evaluated variance homogeneity, and the Mauchly test checked for sphericity. A two-way mixed ANOVA with repeated measures was then used to examine the time × group interaction for all variables. When significant interactions were identified, post hoc analyses using Bonferroni’s multiple comparisons test were performed to explore differences both within groups (pre vs. post) and between groups (NMG vs. MCG vs. CG). To assess the effect size for the time × group interaction, partial eta squared (ηp^2^) was calculated, with interpretations based on ηp^2^ values of 0.01, 0.06, and 0.14 indicating small, moderate, and large effect sizes, respectively. For multiple comparisons, effect sizes were determined using Cohen’s d, with thresholds set at ≥0.2 for small, ≥0.5 for moderate, and ≥0.8 for large effects [40]. For all analyses, an α value of 0.05 was considered.

## 3. Results

Out of the 48 participants initially assessed, 2 were lost to follow-up (one from NMG and one from MCG). One participant was hospitalized due to pneumonia (NMG), and another withdrew for personal reasons following the death of a husband (MCG) (Figure 3). Despite these dropouts, an intention-to-treat analysis was conducted using a single imputation with a LOCF method. No negative side effects, injuries, or harm was reported with the intervention received by participants.

Table 1 presents the baseline characteristics of participants, showing no significant differences in age, body weight, height, or BMI across the groups (*p* ≥ 0.05).

Table 2 shows the interactions between the variables of postural balance and COP displacement in the OE and CE conditions of the intervention groups. Significant time × group interactions were observed in the CE condition. These interactions were presented in mean radius (F = 0.984; *p* = 0.017; ηp^2^ = 0.184), AP velocity with CE (F = 10.01; *p* = 0.001; ηp^2^ = 0.312) and ML velocity with CE (F = 4.30; *p* = 0.027; ηp^2^ = 0.163). Multiple comparisons showed significant differences between the pre- and post-assessment in the NMG for the COP variables: mean radius with CE (*p* = <0.001; ES = 0.74), AP velocity with CE (*p* = <0.001; ES = 1.05), and ML velocity with CE (*p* = 0.029; ES = 0.80). Similarly, significant differences were also found in AP velocity with CE (*p* = 0.002; ES = 0.98) and ML velocity with CE (*p* = 0.035; ES = 1.10) in the MCG. The CG showed no significant changes after 12 weeks of the intervention. Intergroup comparisons also revealed no significant differences between participants who performed the three types of training (*p* ≥ 0.05).

For the functional performance tests, significant time × group interactions were present in the overall score of the SPPB test (F = 11.49; *p* = <0.001; ηp^2^ = 0.343), gait speed subtest (F = 5.513; *p* = 0.012; ηp^2^ = 0.198), and stand-sit test (F = 5.731; *p* = 0.010; ηp^2^ = 0.206). There was no significant interaction in the balance score (F = 2.099; *p* = 0.148; ηp^2^ = 0.087) (Table 3). Multiple comparisons revealed significant differences in the SPPB test score between pre- and post-assessment for both the MCG (*p* = 0.004; ES = 0.09) and the NMG (*p* < 0.001; ES = 1.38), with the effect being greater in the latter (Figure 3). Regarding the subtests that make up the SPPB, significant changes were only observed after 12 weeks of intervention in the NMG participants in the gait speed test (*p* = 0.003; ES = 0.76) and the sit-to-stand chair test (*p* = <0.001; ES = 1.19). When performing the intergroup analysis, significant differences were evident between the NMG and CG groups in the overall SPPB test score (*p* = 0.003; ES = 1.06). Finally, muscle strength, which was assessed by the handgrip test, did not show significant time × group interactions (F = 3.216; *p* = 0.061; ηp^2^ = 0.127), and when performing intragroup and intergroup comparisons, no significant differences were observed before and after the interventions.

## 4. Discussion

This study demonstrated improvements in postural balance variables (i.e., AP and ML velocity) only in the CE condition and in functional performance among participants who underwent neuromuscular and multicomponent training. No significant differences were observed in muscle strength, as measured by the handgrip test, after 12 weeks. Although both training modalities improved the study variables, the effect sizes indicated that the most significant changes occurred in participants who completed multicomponent training combined with 12 weeks of neuromuscular training. These results suggest that this combined approach significantly enhances the benefits of multicomponent training, as has been widely reported in national and international literature.

Previous studies have reported that older women with osteoporosis who performed neuromuscular training demonstrated improved postural balance variables [41]. Similar results were reported by Ahmad as well as Rezaeipour and Apanasenko, who showed that after 8 and 6 weeks of intervention through a neuromuscular training program, ML and AP velocity improved in healthy older adults and those with diabetic neuropathy, respectively [26,42]. With aging, oscillations in postural balance variables increase due to the adaptive response caused by a reduced capacity for sensory integration [43]. A neuromuscular training program can provide adequate information to peripheral sensory receptors, specifically mechanoreceptors, so that the integration of muscle responses is more efficient [23]. Therefore, a possible explanation for our results is that the improvements induced by this type of training could reduce the need for older adults to generate compensatory oscillations, resulting in better postural balance.

On the other hand, in this research, improvements in postural balance were mainly evidenced in the CE condition. Since the evidence indicates that neuromuscular training improves the quantity and density of mechanoreceptors, it could be hypothesized that an increase in their number promotes better postural response and reduces oscillations without the need for visual information [44,45]. Multicomponent training also improved postural balance variables, which could be attributed to the lower body muscle strengthening and balancing activities, which favor the control of AP and ML oscillations [46,47].

Another finding of this research was that the individuals who participated in the neuromuscular training program showed improvements in functional performance, evidenced by better scores on the SPPB test and each of its subtests, particularly in the four-meter gait speed and the five times sit-to-stand test. Similarly, those who only performed multicomponent training also experienced improvements in functional performance scores. Although both groups showed improvements in this variable, the effect sizes were greater in the group that included neuromuscular training, suggesting that this type of training could have an “enhancing” effect compared to conventional and isolated modalities.

Improvements in functional performance of older adults who engage in multicomponent training have been widely documented [48,49,50,51] and are consistent with the findings of this study. However, fewer studies have shown the effects of neuromuscular training on this variable. A recent systematic review led by Concha-Cisternas et al. [52] showed that neuromuscular training has beneficial effects on functional performance in the older population. Of the ten articles included in the review, six reported significant changes in favor of those who underwent this type of training [52].

Among the tests that assessed functional performance, gait speed significantly improved in those who underwent neuromuscular training. These findings are consistent with previous reports in the older population, where participating in such interventions for 8 weeks improved spatiotemporal gait parameters such as speed, step length, and cadence [53]. One possible explanation for these improvements is that neuromuscular training enhances somatic and proprioceptive responses throughout the body, allowing for more intense activation of stabilizing muscles and, in turn, stimulating postural control centers. This leads to more efficient performance in activities involving agility and dynamic balance, such as walking [53,54].

Another test included in the SPPB to assess functional performance that significantly improved in participants who underwent neuromuscular training was the five times sit-to-stand test. Resende Neto et al. [54], in a randomized controlled trial involving 32 healthy older adults, reported improvements in muscle strength and test performance after neuromuscular training compared to a control group, showing similar findings to those reported in this research. A hypothesis that may explain these improvements is that the strength exercises included in neuromuscular training induce neural adaptations, which promote adequate and coordinated intramuscular and intermuscular contractions, resulting in better motor performance [55,56,57].

Finally, muscle strength assessed with the handgrip test showed an increase in both groups after 12 weeks of intervention; however, this improvement was not significant. This could be explained by the fact that the training modalities in this study included few specific activities targeting the forearm, wrist, and hand musculature. Despite this, an increase in grip strength was observed, which is a relevant finding given that this variable is currently considered a strong indicator of overall strength and muscle mass in older adults [58,59].

### Strengths and Limitations

One of the key strengths of this study lies in its double-blind, randomized controlled design, with concealed allocation and an intention-to-treat analysis. This design minimizes bias and ensures that the results are robust and reliable. As well, this study included the implementation of validated instruments to measure the variables.

Among the limitations, it should be noted that the assessors were not blinded, as the study was double-blind rather than triple blind. While participants and therapists were unaware of the intervention details, the assessors were not, which could introduce bias. Future studies should consider blinding the assessors to mitigate this risk. Additionally, the exclusion of male participants limits the generalizability of the findings and hinders a broader comparison across different sexes and groups. Another limitation was that the study did not record the physical activity of the control group, but participants were given strict instructions to maintain their daily activities regularly (which did not include physical activity, according to the eligibility criteria). Lastly, there was no follow-up of clinical trial participants to assess the medium- and long-term effects of the training programs.

## 5. Conclusions

The addition of neuromuscular training to a multicomponent training program improved postural balance variables and functional performance in older women. These findings could have an impact on clinical practices, as they support the implementation of neuromuscular training programs in geriatric rehabilitation, contributing to a reduction in fall risk and promoting greater independence in this vulnerable population. Additionally, these results could guide future research in this area, highlighting the need for further studies to explore the optimization of training programs, their ideal duration, and the inclusion of various exercise modalities to maximize benefits across different age groups and levels of physical ability. Therefore, based on these findings, we suggest implementing and incorporating this program into interventions aimed at the older population.

## Figures and Tables

**Figure 1 jfmk-09-00195-f001:**
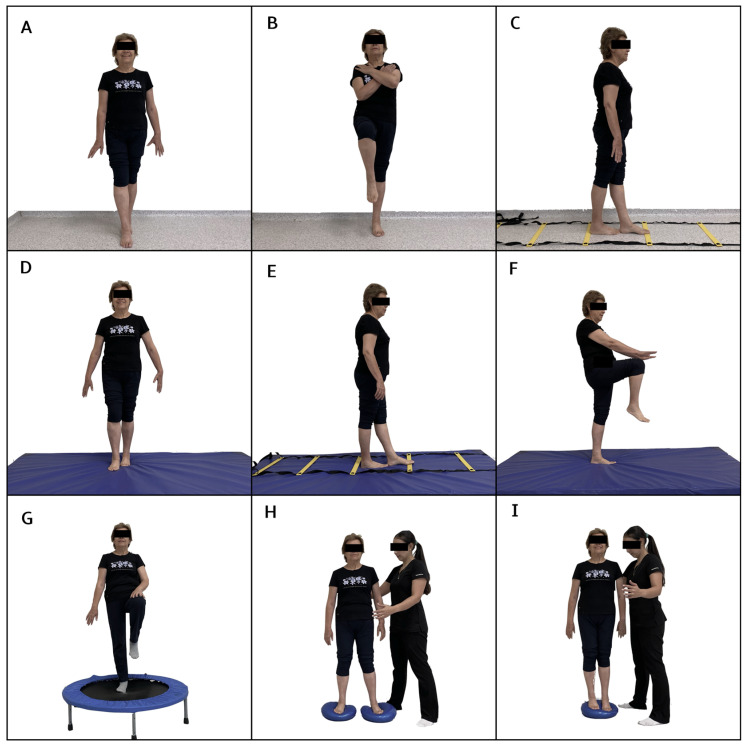
Exercises included in the neuromuscular training program for the lower limb. (**A**,**B**): exercises performed on a stable surface with eyes open, incorporating various foot positions (separated, together, semi-tandem, and tandem). (**C**): walking in a straight line. (**D**,**E**): exercises adapted to an unstable surface. (**F**): static walking with knee elevation. (**G**): exercises on unstable surfaces with eyes open and eyes closed. (**H**,**I**): Exercise on an unstable surface with reduced supports.

**Figure 2 jfmk-09-00195-f002:**
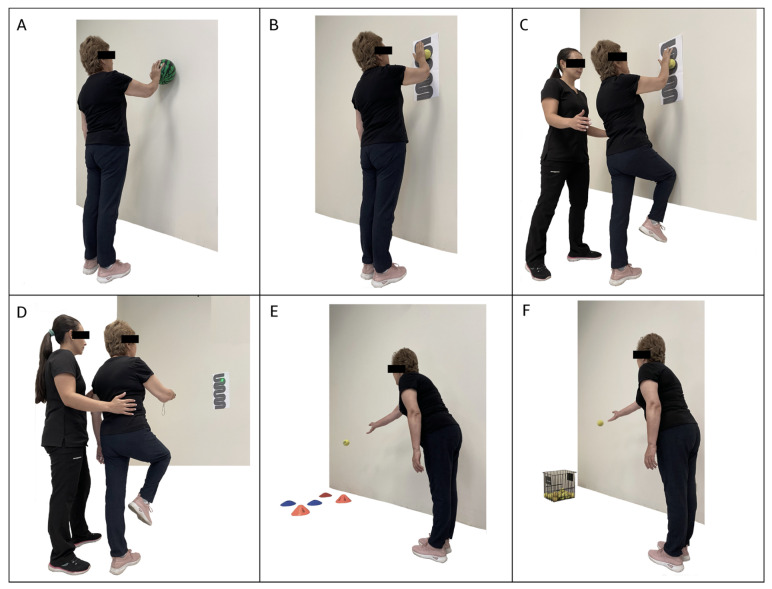
Exercises included in the neuromuscular training program for the upper limb. (**A**): circles of different sizes on the wall. (**B**): wall tracings with increased complexity and smaller ball. (**C**,**D**): tracing patterns on the wall with balls while in a single-leg stance and tracing from a greater distance using a laser pointer. (**E**,**F**): Throwing balls of different sizes with increased difficulty as throwing distances increase.

**Figure 3 jfmk-09-00195-f003:**
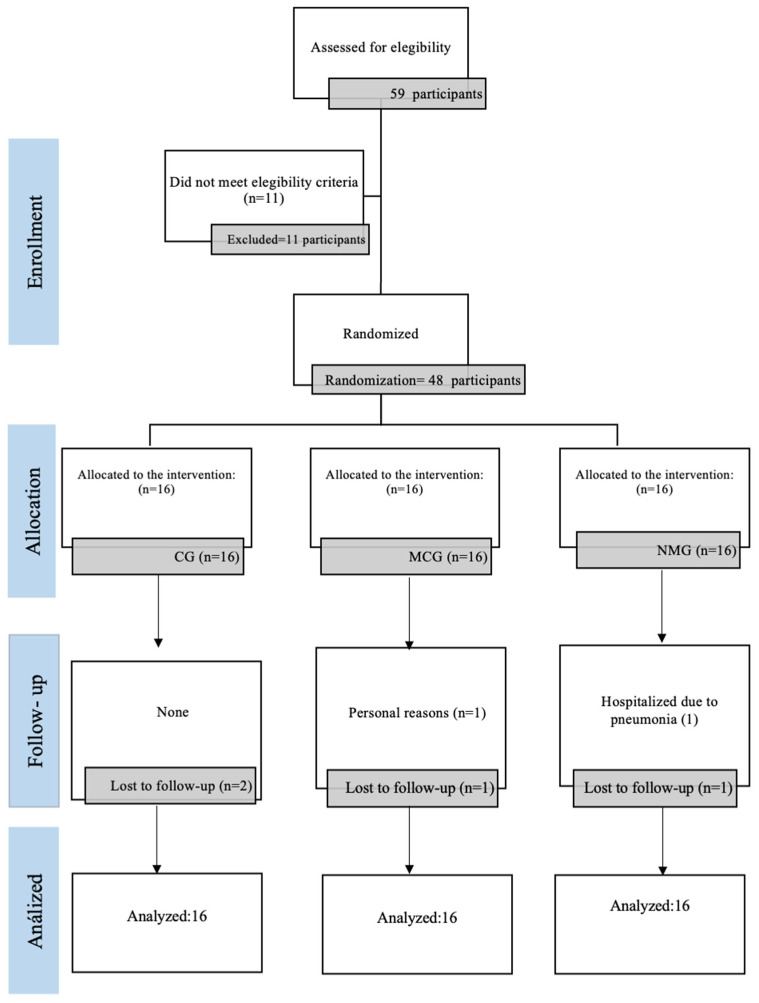
CONSORT flowchart.

**Table 1 jfmk-09-00195-t001:** Baseline characteristics of the sample.

	NMG	MCG	CG
	(n = 16)	(n = 16)	(n = 16)
Age (years)	71.9 (68.5–75.4)	72.6 (70.0–75.5)	71.2 (67.5–74.4)
Body weight (kg)	66.8 (62.5–70.8)	72.4 (65.4–78.8)	70.1 (65.1–76.2)
Height (m)	1.49 (1.47–1.51)	1.53 (1.49–1.57)	1.50 (1.47–1.53)
BMI (kg/m^2^)	29.5 (28.1–31.3)	30.3 (28.2–33.3)	31.0 (28.1–33.5)

Data expressed as means and 95% confidence intervals. BMI = body mass index; NMG = neuromuscular group; MCG = multicomponent group; CG = control group.

**Table 2 jfmk-09-00195-t002:** Time × group interactions of postural balance variables in OE and CE conditions.

COP Variables	Group	Pre-Intervention	Post-Intervention	Time × Group *p* Value	Time × Group F Value	ηp^2^
Area with OE (cm^2^)	NMG	0.011 (0.007–0.015)	0.011 (0.008–0.015)	0.970	0.02995	0.001
	MCG	0.012 (0.008–0.016)	0.011 (0.007–0.016)			
	CG	0.011 (0.007–0.016)	0.010 (0.006–0.015)			
Mean radius with OE (cm)	NMG	0.051 (0.044–0.057)	0.050 (0.039–0.061)	0.368	0.051	0.031
	MCG	0.054 (0.044–0.064)	0.058 (0.046–0.070)			
	CG	0.053 (0.045–0.061)	0.059 (0.048–0.070)			
Mean velocity with OE (cm/s)	NMG	0.237 (0.231–0.244)	0.227 (0.206–0.248)	0.399	0.9614	0.041
	MCG	0.245 (0.228–0.261)	0.235 (0.226–0.243)			
	CG	0.237 (0.231–0.244)	0.242 (0.260–0.224)			
AP velocity with OE (cm/s)	NMG	0.451 (0.398–0.504)	0.378 (0.337–0.420)	0.087	2.757	0.111
	MCG	0.484 (0.417–0.551)	0.469 (0.403–0.535)			
	CG	0.449 (0.370–0.529)	0.445 (0.353–0.537)			
ML velocity with OE (cm/s)	NMG	0.296 (0.267–0.324)	0.272 (0.242–0.320)	0.365	1.059	0.045
	MCG	0.306 (0.279–0.332)	0.291 (0.246–0.336)			
	CG	0.298 (0.249–0.346)	0.297 (0.251–0.342)			
Area with CE (cm^2^)	NMG	0.016 (0.012–0.020)	0.012 (0.008–0.015)	0.088	2.746	0.081
	MCG	0.015 (0.010–0.019)	0.013 (0.008–0.017)			
	CG	0.016 (0.011–0.021)	0.016 (0.012–0.021)			
Mean radius with CE (cm)	NMG	0.065 (0.054–0.075)	0.050 (0.042–0.059)	0.017	0.984	0.184
	MCG	0.062 (0.050–0.074)	0.055 (0.044–0.066)			
	CG	0.067 (0.054–0.079)	0.067 (0.055–0.078)			
Mean velocity With CE (cm/s)	NMG	0.275 (0.253–0.297)	0.249 (0.223–0.275)	0.500	0.716	0.031
	MCG	0.266 (0.247–0.285)	0.252 (0.242–0.261)			
	CG	0.265 (0.251–0.279)	0.256 (0.233–0.280)			
AP velocity with CE (cm/s)	NMG	0.811 (0.630–0.992)	0.511 (0.424–0.597)	0.001	10.010	0.312
	MCG	0.809 (0.652–0.966)	0.570 (0.461–0.678)			
	CG	0.767 (0.552–0.983)	0.801 (0.626–0.977)			
ML velocity with CE (cm/s)	NMG	0.411 (0.331–0.492)	0.307 (0.263–0.351)	0.027	4.307	0.163
	MCG	0.407 (0.333–0.482)	0.296 (0.269–0.322)			
	CG	0.395 (0.308–0.482)	0.411 (0.309–0.513)			

Data presented as means and 95% confidence intervals. ηp^2^: partial eta square; NMG: neuromuscular group; MCG: multicomponent group; CG: control group; ML: mediolateral; AP: anteroposterior; OE: open eyes; CE: closed eyes.

**Table 3 jfmk-09-00195-t003:** Time × group interactions of the functional performance and muscle strength variables.

Physical Performance and Muscle Strength	Group	Pre-Intervention	Post-Intervention	Time × Group *p* Value	Time × Group F Value	ηp^2^
Total SPPB score	NMG	9.50 (8.62–10.37)	11.5 (10.9–12.9)	<0.001	11.49	0.343
	MCG	9.73 (8.85–10.6)	10.9 (10.4–11.4)			
	CG	9.92 (8.98–10.8)	9.92 (9.01–10.8)			
Balance score	NMG	3.33 (2.91–3.75)	3.94 (3.82–4.06)	0.148	2.099	0.087
	MCG	3.20 (2.60–3.79)	3.66 (3.26–4.06)			
	CG	3.28 (2.80–3.76)	3.35 (2.87–3.84)			
Gait speed score	NMG	3.16 (2.64–3.68)	3.77 (3.56–3.99)	0.012	5.513	0.198
	MCG	3.73 (3.48–3.98)	3.93 (3.79–4.07)			
	CG	3.71 (3.44–3.98)	3.57 (3.27–3.86)			
5× Sit-to-stand chair score	NMG	2.72 (2.16–3.28)	3.77 (3.50–4.05)	0.010	5.731	0.206
	MCG	2.80 (2.27–3.32)	3.33 (3.06–3.60)			
	CG	2.92 (2.31–3.54)	3.00 (2.40–3.59)			
Hand grip (kg)	NMG	20.3 (18.0–22.7)	22.0 (19.9–24.2)	0.061	3.216	0.127
	MCG	21.9 (17.9–26.0)	24.6 (19.7–29.6)			
	CG	20.5 (17.5–23.4)	20.7 (18.1–23.2)			

Data presented as mean and 95% confidence intervals. ηp^2^: partial eta square; NMG: neuromuscular group; MCG: multicomponent group; CG: control group. SPPB: Short Physical Performance Battery.

## Data Availability

The datasets used and the data analyzed in this study will be made available upon reasonable request to the corresponding author (E.G.-M.).

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
