# Peer review of "Effects of Neuromuscular Training on Postural Balance and Physical Performance in Older Women: Randomized Controlled Trial"

_jfmk, 2024, doi:10.3390/jfmk9040195_

Round 1
Reviewer 1 Report
Comments and Suggestions for Authors
The paper aims to evaluate the effects of neuromuscular training on postural balance and physical performance in older women through a randomized controlled trial. The main contribution of this study is the demonstration of significant improvements in postural balance and physical performance following neuromuscular training, highlighting its potential as an effective intervention for the elderly population. Strengths include a well-defined study design and appropriate statistical analysis.
General comments
The hypothesis is clear and testable. However, more detailed background information on the specific mechanisms by which neuromuscular training impacts balance and physical performance in older adults could strengthen the manuscript.
There are no major methodological inaccuracies, but some areas could be further clarified:
The description of the randomization process (line 106) could include more detail on the blocking strategy used.
The training protocols for both the multicomponent and neuromuscular groups should be detailed more comprehensively, including specific exercises, intensity, and progression strategies.
The references are generally appropriate, but the manuscript could benefit from additional citations of recent studies (within the last 5 years) to support the statements made, especially in the introduction and discussion sections.
Specific comments
Line 2: Change "older woman" to "older women" to correct plural usage.
Line 5: Add a comma after "Eduardo Guzmán-Muñoz1,10*" for consistency in the author list format.
Line 21: Correct the hyphenation error by changing "so- matosensory" to "somatosensory."
Line 23-25: The introduction mentions morphological and physiological changes but lacks citations to support these claims. Consider adding references to support these statements.
Line 26: Change "adults" to "women" to specify the demographic focused on in the study.
Line 26-27: The sentence "The randomized controlled trial included 48 participants aged 65-80, divided into three groups: a control group (CG), a multicomponent training group (MCG), and a neuromuscular training group (NMG)" should include more details on how participants were randomized into these groups and the rationale behind the age range of 65-80.
Line 33-35: When mentioning significant time×group interactions, include effect sizes or confidence intervals to provide a clearer interpretation of the findings.
Line 36: Ensure consistent hyphenation of terms like "sit-to-stand." Here it is hyphenated, but it appears unhyphenated elsewhere in the manuscript.
Line 75: Change "times" to "times," to maintain consistent formatting of numerical expressions.
Line 103-107: Clarify the randomization process described as "double-blind randomized controlled clinical trial." Specify the blocking strategy and how blinding was maintained for both participants and physical therapists.
Line 107: Remove extra space in "research randomization website was used for randomization (https://www.randomizer.org)."
Line 120: Rephrase for clarity: "Out of these were excluded due to failing to meet the eligibility criteria." Suggestion: "Out of these, some were excluded due to failing to meet the eligibility criteria."
Line 121-122: Provide a clear flow of participant recruitment, inclusion, and exclusion. It is mentioned that 48 participants were randomly assigned, but there is no mention of how many participants were initially considered and excluded.
Line 131: Consider changing "participants diagnosed with vestibular disorders by a healthcare professional" to "participants diagnosed with vestibular disorders by healthcare professionals" to ensure consistency in plural usage.
Line 140: Clarify the statement that "participants were unaware of their group assignments." Specify how this was ensured and if there were any challenges in maintaining blinding, especially given the nature of the interventions.
Line 146: Provide more detail on how postural balance was assessed using the force platform. Include specifics about the protocol and conditions (e.g., duration of trials, number of repetitions).
Line 162: Change "velocity mean" to "mean velocity" for consistency with standard scientific terminology.
Line 180: When describing muscle strength assessment, specify if any calibration or standardization procedures were followed for the dynamometer to ensure accurate measurements.
Line 184: Modify for clarity: Change "elbows flexed at 90º and forearms and wrists in a neutral position." to "elbows flexed at 90º, with forearms and wrists in a neutral position."
Line 210-215: Provide a more detailed description of the multicomponent and neuromuscular training programs, including specific exercises, intensity, and progression strategies over the 12-week period.
Line 211: Correct the sentence structure: "The neuromuscular training program was added to the multicomponent training for 12 weeks. distributed in 2 interventions per week on non-consecutive days totaling 24 sessions." Suggestion: "The neuromuscular training program was added to the multicomponent training for 12 weeks, distributed in 2 sessions per week on non-consecutive days, totaling 24 sessions."
Line 260-273: In the statistical analysis section, include how missing data were handled and whether any sensitivity analyses were performed to assess the robustness of the results.
Line 275: The word "an" should be removed in "an intention-to-treat analysis was conducted," as it reads "Despite these dropouts, an intention-to-treat analysis was conducted."
Line 275-278: The intention-to-treat analysis is mentioned briefly. Provide more detail on how this analysis was conducted and its implications for the study results.
Line 285-324: In the presentation of results, ensure all p-values and confidence intervals are reported consistently. Some results lack this information, which is crucial for interpreting statistical significance.
Line 290: Ensure consistent formatting in "Body weight (Kg)" to "Body weight (kg)" for standard scientific unit formatting.
Line 326: Add a space after the comma in "post-intervention,with" for correct punctuation.
Line 338-341: The discussion section should better integrate the study findings with the existing literature. Include a comparison of the results with similar studies to provide context and highlight the study's contributions.
Line 355-358: When discussing the implications of the study, consider including potential limitations related to the study population (e.g., the exclusion of males) and the generalizability of the findings.
Line 359: In "Participants attended two 90-minute sessions per week on non-consecutive days over a 12-week period, totaling 24 sessions," the phrase "Participants attended two 90-minute sessions per week on non-consecutive days" is repeated twice. Remove the redundant sentence.
Line 406: Change "older woman" to "older women" to correct singular vs. plural form consistency.
Line 435: The term "significantly" might be more impactful in place of "greatly," which appears in "The results indicate that balance improved greatly in the older adults."
Line 506-510: The conclusion section can be expanded to provide more emphasis on the practical applications and potential future directions of this research. Include more on how the findings could influence clinical practices or guide future research in this area.
Line 520-525: In the ethical considerations, provide more information about the ethical approval process and the measures taken to ensure participants' safety and confidentiality throughout the study.
Comments on the Quality of English LanguageThe quality of English in the manuscript is generally good, but minor editing is required to enhance clarity and consistency.
Reviewer 2 Report
Comments and Suggestions for Authors
Interesting study showing that the more work is done, the better the results. Neuromuscular training was not a “different” intervention, but an added intervention, so it is normal to find improvements in the group that worked more.
Therefore, the conclusion of the study is entirely accurate, although very predictable.
Reviewer 3 Report
Comments and Suggestions for Authors
The authors examined the effects of neuromuscular training program on postural balance, physical performance, and muscle strength in older Chilean adults. An RCT with 48 participants 65-80 years of age and three arms (Wait-list Control: CG; multicomponent training: MCG; and neuromuscular training: NMG) was performed. Findings suggest that neuromuscular training is a promising intervention to mitigate declines in balance and physical function. While findings are of interest a few questions remain.
Introduction:
- What is the rationale for the use of a multicomponent training program vs. control vs. neuromuscular training program as points of comparison in this study?
- What are the a-priori hypotheses of this study? What is the rationale for each proposed hypothesis?
Methods:
3. Did participants enroll in prior studies? or perform any concurrent training at the same time?
4. What is the rationale for the specific outcome measures?
5. In wait-list control, was amount of physical activity and type of activity performed during 12 week recorded? Was it used to quantify changes in postural balance, physical performance, and muscle strength?
Results:
5. Did the authors consider post-hoc analysis examining contrasts between groups to better understand differences between groups?
Discussion:
6. How were differences between training modalities formally tested? Results provide significant interactions, but no post-hoc test results are provided.
7. What are additional mechanisms by which neuromuscular training can improve postural control, gait, and muscle strength? What is the role of improved short- medium- and long-term latency responses to these measures? What are the elements of the training that may be most associated with improvements?
